# Compression and Bending Properties of Short Carbon Fiber Reinforced Polymers Sandwich Structures Produced via Fused Filament Fabrication Process

**DOI:** 10.3390/polym14142923

**Published:** 2022-07-19

**Authors:** Sebastian Marian Zaharia, Mihai Alin Pop, Lucia-Antoneta Chicos, George Razvan Buican, Camil Lancea, Ionut Stelian Pascariu, Valentin-Marian Stamate

**Affiliations:** 1Department of Manufacturing Engineering, Transilvania University of Brasov, 500036 Brasov, Romania; l.chicos@unitbv.ro (L.-A.C.); buican.george@unitbv.ro (G.R.B.); camil@unitbv.ro (C.L.); ionut.pascariu@student.unitbv.ro (I.S.P.); valentin_s@unitbv.ro (V.-M.S.); 2Department of Materials Science, Transilvania University of Brasov, 500036 Brasov, Romania; mihai.pop@unitbv.ro

**Keywords:** sandwich structures, fused filament fabrication, short carbon fiber, mechanical testing, failure analysis, wing structure

## Abstract

Additive manufacturing, through the process of thermoplastic extrusion of filament, allows the manufacture of complex composite sandwich structures in a short time with low costs. This paper presents the design and fabrication by Fused Filament Fabrication (FFF) of composite sandwich structures with short fibers, having three core types C, Z, and H, followed by mechanical performance testing of the structures for compression and bending in three points. Flatwise compression tests and three-point bending have clearly indicated the superior performance of H-core sandwich structures due to dense core structures. The main modes of failure of composite sandwich structures were analyzed microscopically, highlighting core shear buckling in compression tests and face indentation in three-point bending tests. The strength–mass ratio allowed the identification of the structures with the best performances considering the desire to reduce the mass, so: the H-core sandwich structures showed the best results in compression tests and the C-core sandwich structures in three-point bending tests. The feasibility of the FFF process and the three-point bending test of composite wing sections, which will be used on an unmanned aircraft, have also been demonstrated. The finite element analysis showed the distribution of equivalent stresses and reaction forces for the composite wing sections tested for bending, proving to validate the experimental results.

## 1. Introduction

Composite materials have been developed primarily out of the need to reduce the weight of industrial structures and extend their lifespan, a particularly important goal in aerospace [1], automotive [2], marine [3], and construction industries [4]. There is virtually no field, from cutting-edge industries to traditional industries, to which composite materials have not penetrated. One direction intensively researched and used in industrial applications of composite materials is the sandwich structure of composite materials. They are a special category of composite materials, which are made by attaching two thin but rigid skins (shells) to a light but thick core. The advantages of composite sandwich structures compared to classic metal structures are the greater rigidity, better bending strength, high fatigue strength, low weight, and good thermal and sound insulation. Composite sandwich structures can be made by new manufacturing methods, such as Fused Filament Fabrication (FFF). Composite sandwich structures include a base material and outer skins that can be made from the same material or from a different material than the core. The FFF process benefits from a relatively low cost, a wide range of materials, a high manufacturing speed, and a simple, quiet, and safe process. However, the FFF process also has some disadvantages: the strength of the parts in the direction of 3D printing (Z) is always lower than their strength in the XY plane, poor surface quality, low dimensional accuracy, and specific defects of 3D printing (voids, layer shifting, blocked nozzle, warping, delamination, stringing. Elephant’s foot).

Filaments reinforced with short fibers (chopped) or continuous fibers have been developed in order to improve the mechanical strength of parts manufactured through the FFF process [5]. However, the fabrication by the FFF process with filaments reinforced by composite fibers introduces some problems, such as low adhesion between the layers of material, fiber orientation, poor connection between fiber and a matrix with the formation of voids, deformation of the shape caused by residual stresses that comes from uneven temperature gradients, an uneven distribution of fibers in the fiber-reinforced thermoplastic filament, and surface roughness [6,7,8]. Although 3D-polymer composites reinforced with short fibers show significant performance improvements over pure plastics (Polylactic acid -PLA, Acrylonitrile butadiene styrene—ABS, Polyethylene terephthalate glycol—PETG), the mechanical properties are much lower compared to 3D-printed polymer composites reinforced with continuous fibers. As the fibers have a specific stiffness and a much higher specific strength than the matrix, it is recommended to design composite structures so that the loads are mainly supported and transmitted by the fibers. In order to facilitate this design goal, maintaining fiber continuity is essential [9]. Some researchers [10,11,12] have been dedicated to determining the mechanical performance by varying the manufacturing parameters of the FFF process (infill density, infill pattern, deposition layer height, printing speed, printing temperature, nozzle diameter) for polymer composites reinforced with short fibers. Yasa et al. [13] showed that there was a severe level of anisotropy in the mechanical properties for the modulus of elasticity caused by insufficient adhesion between the layers deposited in the construction direction of the parts made from chopped carbon-reinforced polymers. Another study [14] on nylon filaments reinforced with carbon fiber indicated that the hardness and the tensile strength are influenced by the construction direction of the part, the infill density, and the thermal stresses, while the resilience is influenced only by the construction direction, and the relationship between the mechanical properties and the infill factor is not linear. In a study [15], the effects of the process conditions on the manufacture of polymer composites reinforced with short carbon fiber made by the material extrusion process were investigated at micro and macro levels. Image-based statistical analysis was used for microstructural characterization (by example, fiber volume fraction), and the results obtained confirmed that the manufacturing parameters play an important role in void generation and the distribution of the void volume fraction. Zhong et al. [16] found that the strength of parts made in the FFF process significantly improved by adding short glass fibers to an acrylonitrile–butadiene–styrene (ABS) matrix.

For composites reinforced with continuous fibers, the composite can be 3D printed by a double extrusion [17] or co-extrusion [18] method. Studies have been performed [19,20,21] to compare the performance of polymer composites reinforced with short and continuous carbon fiber manufactured by the FFF process. Araya-Calvo et al. [22] determined the bending and compression performance of polyamide 6 (PA6) reinforced with continuous carbon fiber and studied the effect of fiber pattern, reinforcement distribution, and print orientation on mechanical properties. The study [23] investigated the microstructure and mechanical properties of polyamide composites reinforced with continuous carbon fiber manufactured by the FFF process. The mechanical properties of 3D-printed composite specimens reinforced with continuous fibers indicate high values of rigidity and strength for longitudinal composites, where the fibers have been aligned unidirectionally in the loading direction. Defects specific to the FFF process (high void content, poor interlayer bonding, and inhomogeneous fibers distribution) have a significant impact on the strength to interlaminar shear and on the breaking behavior of 3D-printed specimens [23,24,25,26,27]. However, researchers have studied and significantly improved the mechanical performance of composites reinforced with continuous fibers, using various types of tests as compression [28,29], tensile [30,31,32], bending [33,34,35], fatigue [36,37], and impact [38,39].

Sandwich structures manufactured by the thermoplastic extrusion process represent an intensely researched direction, primarily due to the continuous development of 3D printing systems and filaments. Thus, lately, sandwich structures from various types of materials with different core topologies have been manufactured by the FFF process and tested afterward. Zaharia et al. [40] studied three types of cores (honeycomb, diamond, and corrugated) made by the FFF process from polylactic acid/polyhydroxyalkanoate (PLA/PHA). The results showed good performance for the diamond core in three-point compression and bending, and the corrugated core showed good performance for the tensile stress. In a recent study, 3D printed sandwich structures, using TPU as the base material and ABS, PMMA, and HIPS for coatings, have been tested for: bending and impact [5]. The mechanical properties of sandwich structures manufactured by the FFF process have also been improved by the simultaneous use of reinforced filament materials and by optimizing the parameters of the 3D printing process. Therefore, using the FFF process, hybrid composite laminates were manufactured that showed better mechanical properties [41,42] compared to conventional materials. Zeng et al. [43] analyzed the static performance of continuous fiber reinforced composite trapezoidal corrugated sandwich structures with shape-memory capability manufactured by the FFF process. It was found that the studied structures can offer new opportunities for use in lightweight systems and multifunctional applications through the studied concept of shape recovery. Three-dimensional printed sandwich composites were investigated [44] in order to determine the performance of bending in three points, after which the structural failure modes were analyzed. The result indicates that the addition of a glass micro balloon increases both the specific modulus and the strength of sandwich composites. Galatas et al. [45] investigated the mechanical properties of composite sandwich structures with an ABS core and a carbon-fiber-reinforced polymer coating. The results showed an improvement in the specific strength and modulus of elasticity as the number of polymers reinforced with carbon fiber increased, and the analyzed material was implemented on the structure of a clamp in the structure of a quadcopter. Recently, Andrey et al. [46] manufactured the lattice frame of a small unmanned aerial vehicle from continuous fiber-reinforced composites using the FFF process.

The present paper is aimed to design and test sandwich structures reinforced with short carbon fibers, with various core topologies (Z, C, Hat) fabricated by the FFF process. The performances at flatwise compression and at three-point bending were evaluated for the three core configurations. In this paper, composite wing sections with a C core type spar, manufactured by the FFF process, from an unmanned aerial vehicle (UAV) are studied as an application of short fiber-reinforced composite sandwich structures. The performance evaluation of the three-point mechanical bending of composite wing sections was also determined in this study. Moreover, microstructural analysis of the defects appeared after the tests were performed, starting from the filament, the sandwich structures, and the wing sections. A final stage was dedicated to the comparative study of the wing sections, between the experimental results and the results obtained using simulation by finite element method.

## 2. Materials and Methods

### 2.1. Design of Composite Sandwich Structures

The composite sandwich structures were designed in SolidWorks 2021 (Dassault Systèmes SolidWorks Corporation, Waltham, MA, USA), considering the specific regulations in force (MIL-STD-401) regarding the testing of such specimens. For each type of test (compression and three-point bending), three sandwich structures were designed with the following core types: C core (Figure 1a–c), Z core (Figure 1d–f), and Hat core (Figure 1g–i). The dimensional details regarding the positioning of the three core types can be found in Figure 1. The sandwich specimens used in the compression tests had the following dimensions: length 50 mm, width 50 mm, and height 15 mm. For the three-point bending test specimens, the dimensional characteristics were as follows: length 150 mm; width 20 mm, and height 15 mm.

### 2.2. Design of Wing Sections

The wing sections designed and tested in this study are part of the structure of an unmanned aerial vehicle (UAV) made exclusively by 3D printing composite materials. The wing is the main component of the airplane because around the wing, the whole skeleton of the airplane is built, and the design begins with it. The wing creates, during the movement of the airplane, the lifting force that keeps the airplane in the air. The forces acting on the wing are: aerodynamic forces (lift, drag, and aerodynamic moment) and mass forces that can be concentrated or distributed, coming from various components (own engine weight, landing gear, weight of the wings). These forces will be multiplied by the load factor corresponding to the evolution of the aircraft. The stresses produced by these forces are defined in relation to the line of the centers of rigidity of the wing and generally consist of: a dominant stress—shear–bending in the normal direction of the chord and twisting (normal shear force, bending moment, torque) and a secondary load (shearing–bending in the plane of the chords). The design of the wing sections was carried out in SolidWorks 2021, starting with the coordinates for the NACA 4415 airfoil. The placement of the main spar was performed according to the aeronautical building techniques, as follows: the first spar, a C core type, was positioned around 17–25% of the chord, and the second spar, an X core type, was placed at around 30–45% of the chord, and the third spar, again a C core type, was located between 60–75% of the chord. The internal configuration of the wing section (Figure 2a) had a structure with three spars, as follows a C core type spar on the wing leading edge, an X core type spar which takes the load from the central part of the wing, and a C core type spar that is required to take the loads from the wing trailing edge. The thickness of the wing skin was 1 mm, the width of the section was 43 mm, and the thickness of the three spars was 0.8 mm (Figure 2b).

### 2.3. Manufacture of Sandwich Specimens and Wing Sections Using the FFF Process

#### 2.3.1. Materials

The filament used in the manufacture of the composite sandwich specimens and wing sections was a BASF Ultrafuse PAHT CF15 (Emmen, The Netherlands), with a diameter of 2.85 mm [47]. The commercial filament is composed of a polyamide matrix reinforced with short carbon fibers at a volume fraction of 6.5 ± 0.2% [48]. For the FFF specimen process, the BASF Ultrafuse PAHT CF15 filament was pre-dried at 70 °C for 12 h, according to the manufacturer’s specifications, using the Wanhao Box 2 dryer (Wanhao, Jinhua, China). Packages of silica gel were placed inside the closed enclosure of the 3D printer to keep the absorption of moisture by the polymer as low as possible. The filament was analyzed microscopically before being used in the 3D printing of specimens and wing sections. The filament was analyzed microscopically using Nikon Eclipse MA 100 microscopes (Nikon Corp., Tokyo, Japan) before being used in the FFF process. Voids in the BASF Ultrafuse PAHT CF15 filament were highlighted. These defects are due to the poor incorporation of short fibers into the thermoplastic matrix during the filament manufacturing process [48]. These defects have been reported in other studies [49,50,51] in which short fiber-reinforced filaments were analyzed, where it was observed that they are prone to porosity due to low adhesion between thermoplastic and carbon-fiber surfaces. The following can be seen in the microscopic images of the filament: the white lines (Figure 3a) or the white dots (Figure 3b) represent the carbon fibers, and the darker areas represent voids that come from the filament manufacturing process.

#### 2.3.2. Manufacture of Composite Sandwich Specimens

All of the composite structures were printed using the BCN3D Epsilon W50 3D printer (Barcelona, Spain) from the BASF Ultrafuse PAHT CF15 filament. Positioning (Figure 4) and the 3D printing of the structures were performed on the edge (XZ plane) without support.

The manufacturing parameters of the FFF process for the composite sandwich specimens and wing sections are described in Table 1.

### 2.4. Testing of Composite Sandwich Specimens

The mechanical tests for the composite structures (sandwich structures and wing sections) were performed on the WDW-150S universal testing machine (Jinan Testing Equipment IE Corporation, Jinan, China). For the experimental tests (flatwise compression and three-point bending), five specimens were manufactured by the FFF process for each core configuration (C, Z, and Hat), 15 for flatwise compression and 15 for bending, a total of 30 specimens. The compression-tested specimens (Figure 5a) were manufactured in accordance with ASTM C365 [52], and the loading speed was 5 mm/min. These tests were performed to determine the mechanical performance of composite sandwich structures (flatwise compressive strength and modulus of elasticity for compression). The three-point bending tests (Figure 5b) evaluated the bending strengths of 3D printed composite specimens until failure using a 5 mm/min speed in accordance with ASTM D790 [53]. Five wing section specimens were tested at three-point bending (Figure 5c) at a displacement speed of 5 mm/min. Following the three-point bending tests of the 3D-printed wing sections, the load that the wing structure will be able to support during the flight was determined using the three spars (two spars with a C section type and one spar with an X section type).

## 3. Results and Discussion

### 3.1. Flatwise Compression Performance of Carbon Fiber Sandwich Structures

Five compression tests were performed for each type of core specimen (C, Z, and Hat), 3D printed by the FFF process. From these tests were determined the mean values of compressive strength and modulus of elasticity for compression (Figure 6a). Figure 6b shows a representative load–displacement curve for a compression test for each core type from the composite sandwich structure. The compressive performance of different composite sandwich structures varies depending on the different core topologies. The mechanical performances (compression strength and modulus of elasticity of compression) were calculated with the standardized calculation relationships of the compression tests introduced by the manufacturer in the software of the test equipment. These values of compressive strength and compression modulus are automatically generated in the test report that came out from the testing of the sandwich specimens. The sandwich structures with a C core had the lowest compression performance (5.6 MPa compressive strength and 0.256 Gpa modulus of elasticity for compression), and the composite sandwich structures with a Hat core had the highest performance (14 Mpa compressive strength and 0.44 Gpa modulus of elasticity for compression). The values of the compressive strength of the Hat core sandwich structure are higher compared to the results obtained in other studies for different types of materials manufactured by additive processes, such as the compressive strength of the diamond core sandwich structure (3 Mpa) manufactured by the FFF process from PLA-PHA material [40]; compressive strength of sandwich structures with diamond core (6.97 Mpa) manufactured by the Vat Photopolymerization process [54]. Load–displacement curves typical of compression tests present similar evolutionary patterns, which can be divided into three different stages: elastic deformation, deformation plateau, and densification [55,56,57]. In the flatwise compression test, the plate of the test machine was close to the sandwich specimen but without coming into contact with it. Thus, there was a distance at which the equipment recorded only the movement without loading the specimen. In the case of the composite specimens from this study, they showed similar behavior: an elastic domain, a short transient elastic-plastic domain, and then a progressive failure of the structure to reach the maximum compressive strength. The irreversible failure of the composite sandwich specimens appeared as follows: for specimens with a C core, the maximum compression force was 11.3 kN, for specimens with a Z core, the maximum compression force was 13.2 kN, and for specimens with a Hat core, the maximum compression force was 25.5 kN. Following the compression tests, the maximum displacements were between 1.4 mm and 1.8 mm. The phenomenon of densification could be observed only in the sandwich structures with a Hat core because they have a denser core network but also the highest strength.

The main statistical indicators (Table 2) were determined, for the values of compressive strength and elasticity of compression, of composite sandwich specimens manufactured by the FFF process.

Of all the variables that characterize variations, the standard deviation is used for certain statistical analyzes. However, the standard deviation provides an absolute estimate of the measurement of the dispersion of the values, and, in order to understand how large it is in relation to the values themselves, it is necessary to introduce a relative indicator. This indicator is called the coefficient of variation (CV) and is defined as the ratio between the standard deviation and the mean value of the data sample, expressed as a percentage. Moreover, in the statistical analysis of the experimental data, there is a system of indicators that reflect the homogeneity of the data and the stability of the processes, including the coefficient of variation (CV). In the case of data obtained from compression tests, it can be stated that the data are homogeneous, and the averages are representative for the six sets of values because the coefficient of variation is close to zero (CV < 30%).

The compression load was mainly supported by the three core configurations, and the role of the skin is to take over and transmit the loads to the core. The core with C configuration (Figure 7a) showed, during the compression test, a shear buckling of the core, which determined its rupture. Another mode of failure that occurred after the compression testing of this type of specimen was cracking, followed by a debonding between the skin layers and the C core. Microscopic inspection of a defective Z core sandwich specimen (Figure 7b) seems to confirm the complexity of the deformation mechanism, in which the typical buckling modes of the core appear at the neighboring Z profiles, together with the cracking and rupture of both the core and the skin. Shear buckling shows a mode of failure specific to the configuration of the Hat core (Figure 7c), with a deformation directed to the inside of the structure, totally opposite to the other two structures (with C core and Z core) analyzed above. This structure is much more rigid, the two components (core and skin) are well designed, and the phenomenon of core densification has led to a higher strength. Due to the denser core, the Hat core sandwich skin did not crack; only the core failed.

### 3.2. Three-Point Bending Behavior of Carbon Fiber Sandwich Structures

Following the three-point bending tests of the composite specimens, for each type of C, Z, and Hat core specimen, the mean values of bending strength and modulus of bending elasticity were determined (Figure 8a). The bending tests were performed on a number of five specimens for each type of core C, Z, and Hat, and with the help of experimental data were obtained and exposed the representative curves load–displacement. C core sandwich structures and Z core sandwich structures showed close values for the mean bending strength, 12.4 MPa for C core, and 12.2 Mpa for Z core. In contrast, Hat core sandwich structures have approximately 50% higher strength compared to the other two composite structures. The superior mechanical performance of Hat core sandwich structures is due to the denser structure of the core, corroborated with the occurrence of its densification phenomenon. Figure 8b illustrates the evolution of the force as a function of displacement for composite sandwich specimens, and the mechanical behavior has two areas:In the first domain, a linear elastic behavior of composite sandwich structures was observed. At the beginning of this domain, for Hat core specimens, the force increases and corresponds to a smaller displacement. This clearly demonstrates that the bending stiffness of this sandwich structure is enhanced by the shear stiffness of the Hat core;The second domain comprises the final range of the curve and corresponds to the nonlinear behavior of the material until the sudden rupture of the composite sandwich specimens. The end of this area highlights the failure mode of the composite sandwich structures. The core is shear loaded, and its failure occurs as the critical value (shear strength) of the core material is reached by the maximum shear stress.

The irreversible failure of the composite sandwich specimens with the highest bending performance (Hat core sandwich specimens) occurred at a force of approximately 0.66 kN correlated with a maximum displacement of approximately 7.6 mm. The C core and Z core sandwich structures exhibited similar behavior in terms of bending stress according to the Load–Displacement curves. Higher displacement is due to the fact that these two types of core topologies, C and Z, are more elastic with much more distance between the cores compared to the Hat core, which has very close stiffening elements (can be assimilated with a structure with a double core).

The bending performance of Hat core composite sandwich structures is higher or similar compared to the results obtained in other studies for different 3D printed core topologies with various types of materials, such as: the bending strength of the diamond core sandwich structure (16 MPa) 3D printed from PLA-PHA material [40]; the bending stresses for specimens with a core gyroid thickness of 0.75 mm and a CFRP insert diameter of 1.20 mm was 2.3 Mpa, the specimens obtained using the technique of light-curing acrylic resin stereolithography [58]; the value of the bending strength of some gyroid-structured core specimens, manufactured by the FFF process, from wood/PLA filaments, was 11.82 Mpa [59].

For the composite sandwich specimens manufactured through the FFF process, the basic statistical indicators for the values of the bending strength and the modulus of elasticity at bending were calculated (Table 3). Suppose the coefficient of variation (CV) is close to zero (CV < 30%), then the statistically processed data are homogeneous, and the calculated mean is representative of these experimental data sets. The values of the coefficient of variation in the bending tests, were between 1.66% and 9.01%, so it can be concluded that the data are homogeneous.

Different modes of failure have been observed for composite sandwich specimens under bending tests. The C core and Z core sandwich structures have a high degree of flexibility, and the failure mode for the two is similar, namely a stable, progressive defect with local indentation of the skin, both in the loading punch area and in supporting pins. The local indentation of the composite sandwich structures started with an initial phase of linear elastic deformation, followed by a deep local deformation both in the area where the stress is applied (Figure 9a) but also in the area where the specimen surface is in contact with the supporting pins (Figure 9b). This local defect in the composite sandwich structure quickly reduced its load-bearing capacity, resulting in a sudden drop in load required to continue the bending test. For the Hat core sandwich specimens (Figure 9c), there was also a ductile rupture of the skin, which is caused by tangential stresses and is preceded by large plastic deformations. When the structure of the core is sufficiently well dimensioned, the indentation of the skin takes place, followed by the failure by yielding the upper face of the skin, and finally, the structure fails by yielding the lower skin. It is also possible to observe in this type of structure a debonding and cracking of the Hat core from the lower skin.

### 3.3. Bending Performance of Wing Sections

In this study, starting from the sandwich structures analyzed previously, the research on the manufacture and testing of the performance of some wing sections was extended. Thus, the three-point bend test wing was manufactured using three spars (two with a C profile and the main spar with an X profile). Figure 10a describes the results of the bending strengths of the wing specimens, and their values are between 5.5 MPa and 6.5 Mpa. The main statistical indicators were determined for the bending results of the wing sections. Thus, the value of the coefficient of variation is 9.8%, and the mean is 5.8 Mpa; it demonstrates that the data are homogeneous, and the mean is representative. The representative Load–Displacement curve (Figure 10b) shows two main zones: a linear increase, between the applied force and the displacement, with a certain non-linear behavior, towards the maximum of the curve and then a sudden decrease, at the maximum force, at the moment the specimens suffer breakage. It can be observed that the maximum force, until the appearance of irreversible failure in the material of the composite sandwich structure, was approximately 0.3 kN. Furthermore, the maximum displacement at which the irreversible failure occurred in the material of composite sandwich structures was 19 mm.

The five wing section specimens had as a main mode of failure the deformation (indentation) in the loading area, as well as in the fixing area of the two supports. However, two of the wing sections manufactured by the FFF process also showed a complete fracture of the skin (Figure 11a) and in the area where the force was applied. In Figure 11b, there is a top view of the wing skin, where the wing failure can be seen, followed by a crack between the layers of the extruded material.

### 3.4. Microscopic Analysis of Composite Sandwich Structures

Microstructural analyzes were performed for the composite sandwich specimens to highlight the main defects specific to the FFF process for composite filaments reinforced with short carbon fiber. The composite sandwich specimens were prepared for microscopic analysis as follows: the specimens were cut in cross-section, incorporated into the resin, and polished using a granulation suspension of 1 μm and 0.5 μm Al2O3. The composite sandwich specimens were analyzed microscopically on the sections perpendicular to the XY plane (cross section) and according to the construction direction—Z axis (longitudinal section) with a magnification of 100×. On the microscopic analysis, typical defects of short fiber-reinforced composites were found, similar to the ones in other studies [7,23,30,60,61,62]. The defects are rectangular and triangular voids formation, inter-layer voids, voids found in the matrix or filament, and poor adhesion between the fiber and the matrix. Figure 12a shows a 90° corner area of a sandwich structure manufactured by the FFF process, and in this corner, we can see a change in the orientation of the carbon fibers with some voids where the direction of 3D printing had changed. Figure 12b analyzed an area with a partial area of the sandwich structure skin, where the breaking of carbon fibers and pull-out fibers can be observed. Figure 12c shows an area within the core of profile C where the orientation of the carbon fibers and some voids can be seen. The stratified structure of the composite sandwich specimens obtained from the FFF process is visible by microscopic analysis due to interlayer voids and irregular distribution of carbon fibers and matrix (Figure 12d). A solution for better adhesion between layers of extruded carbon fiber material is 3D printing with a lower layer height, which will allow better interlayer adhesion and stability between individual lines [63,64].

### 3.5. Analysis of the Specific Strength-to-Mass Ratio of Composite Specimens

The specific strength-to-mass ratio was used to analyze the mechanical performance of composite sandwich structures. All of the composite sandwich specimens were weighed (Figure 13a), and the mean of the two characteristics (strength and mass) was used to calculate the specific ratio. Thus, the strength-to-mass ratio was evaluated for the three-point compression, and bending tests were conducted on the composite sandwich structures manufactured with the three core configurations (C, Z, H). Following the analysis of this report, the following conclusions can be drawn:For compression tests, H core composite sandwich structures have the highest value; it turns out that this structure can be used for aeronautical components whose main requirement is compression. It can also be seen that the specimens with the Z-core sandwich structure have a higher ratio compared to the C-core structures. This is due to the fact that the Z-core, through the flanges positioned to the left and right of the core, absorbs the compression force much better.For the three-point bending tests, the sandwich structures showed a very close strength-to-mass ratio. However, C-core sandwich structures showed the highest strength-to-mass ratio, as confirmed by the frequent use of this structure in the wing airframes for small aircraft and unmanned aerial vehicles.

### 3.6. Bending Finite Element Analysis of Composite Wing Sections

Finite element analysis of the wing sections manufactured by the FFF process was performed in the static module of the ANSYS 2021 R2 software system (ANSYS, Inc., Canonsburg, PA, USA). For this analysis, the model is consistent with the experimental model of the wing section, and the three-point bending test conditions (section size, PAHT-CF15 material properties, punch and support radius, and the distance between the supports were established in accordance with the three-point bending tests). The finite element analysis at three-point bending was performed following two aspects: the comparative analysis between the defective behavior of the wing sections, tested at three-point bending and the result obtained from the finite element analysis of the same models; the comparative study of the maximum forces that appeared at the breaking of the wing sections, tested at the bending in three points, and the reaction forces, appeared in the structure of the supports, from the model with finite elements. In the finite element analysis, the boundary conditions of the frictionless support type (Figure 14a) for the wing section were established so as not to rotate during the simulation. The displacement of the punch was applied in the middle of the wing section, having a value of 5 mm/min (Figure 14a), under the same conditions as in the case of experimental tests. For the components of the test machine, rigid body properties (which do not deform under the action of forces) were assigned, and for the two supports, the option of a body–ground fixed joint was used (Figure 14b). The discretization of the wing model for finite element analysis at the three-point bending stress was performed with Hexa-type three-dimensional elements with a discretization element size of 0.5 mm (Figure 14c). The punch and the two supports were discretized with the same Hexa element type with a discretization element size of 2 mm (Figure 14c). The friction between the punch, supports, and the surface of the wing section was taken into account, and the coefficient of friction was 0.1 [65].

In the finite element analysis, the elastic-plastic model was used for the wing sections, both for the two skins and for the spar. The finite element model was analyzed according to the properties of the wing sections, showing an infill density of 100%. In the finite element analysis of the wing sections, some simplifying hypotheses were established, and used in other studies [60,66,67,68]: the constituents have a linear elastic behavior, the matrix (polyamide) has isotropic properties; short micro-carbon fibers are transverse isotropic; the fiber matrix has a perfect adhesion and has no voids or defects. The value of the modulus of elasticity was 8386 MPa, assumed to be that of the composite filament, and the Poisson’s ratio had the value of 0.3 [66,68]. The material used in the FEA analysis was described based on the material characteristics provided by the filament manufacturer. The failure criterion used in the finite element analysis, performed using Ansys software, of the wing sections that were printed through the FFF process was Maximum Stress. 

Thus, following the investigation of the wing sections subjected to three-point bending tests (Figure 14d) and the finite element analysis (Figure 14f), it can be seen that the rupture occurs, in both cases, at the upper skin of the wing section, where the equivalent Von Mises stress is maximum (66 MPa). It can also be observed that the entire composite wing structure, tested at three-point bending and finite element simulation, showed the same mode of deformation, namely: a local buckling of the skin in the area where the load is applied. This mode of failure indicates a good behavior of both the wing skin and the network of stiffening structures (the two spars with C cores and the main spar with an X core). Regarding the comparative study, between the maximum forces, which appeared in the three-point bending tests of the wing sections and the reaction forces, which appeared in the supports of the finite element model, it can be stated that there is an adequate validation of the results. If the two values of the forces appearing in the supports are analyzed (Figure 14e) between the experimental and the simulated results, it can be specified that there is an error that falls to a maximum of 3%. This relative error is accepted in the field of aviation and normally occurs due to the simplifying assumptions set out above. Based on the results of three-point mechanical bending tests of the wing sections, it can be established that the data from the computer-generated bending tests (FEA) is an effective method to characterize the deformation behavior of composite wing specimens manufactured by the FFF process.

## 4. Conclusions

Composite sandwich structures are successfully used in the aerospace field for various structural components, such as wing leading edges, fuselage structures, internal wing configurations and feathers, and helicopter blades. In this study, short carbon-fiber composite sandwich structures with three different core topologies, C, Z, and H, were designed, manufactured by the FFF process, and afterward tested.

The compression tests of the three types of specimens showed that the H-core sandwich structures presented the highest compressive strength due to the dense core structure. The analysis of the composite sandwich specimens tested at compression revealed a mode of failure specific to these types of structures, namely core shear buckling, followed by the rupture of the skin and cracks between the layers of extruded material. 

The three-point bending tests showed that the H-core specimens had the highest performance, and the other two had a very close bending strength value. Microscopic analysis of the composite sandwich specimens indicated a face indentation for the C and Z core topologies and for the H core, a failure of both the skin and the core. After studying the mass strength ratio, the following conclusions can be drawn: in the compression test, the H-core sandwich structures showed the best performance; In contrast, in three-point bending testing, C-core sandwich structures performed best.

The demonstration of the feasibility of manufacturing FFF sections of wing sections for an unmanned aircraft using the C core topology was also analyzed in this study. From the three-point bending tests, it can be seen that the wing section can take on approximately 0.3 kN, which is an adequate value for an unmanned aircraft, without the need for other types of structural elements (spars, stringers, ribs). Finite element analysis complements the experimental results for the wing sections and successfully validates the three-point bending tests, with a low relative error value of about 3%. The results of this study provide insights into the development of composite sandwich structures manufactured by the FFF process, for a wide range of engineering applications, especially in aviation, where lightweight components with high structural performance are required.

## Figures and Tables

**Figure 1 polymers-14-02923-f001:**
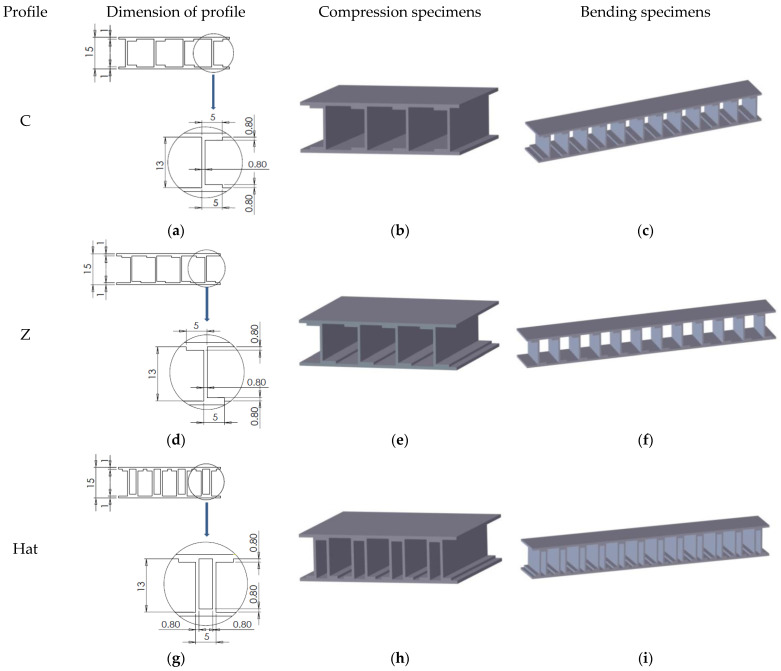
Design and dimensional description of composite sandwich structures (mm): (**a**) C profile dimensions; (**b**) C-core composite sandwich structure subjected to compression test; (**c**) the C-core composite sandwich structure subjected to bending test; (**d**) the dimensions of the Z profile; (**e**) Z-core composite sandwich structure subjected to compression test; (**f**) Z-core composite sandwich structure subjected to bending test; (**g**) Hat profile dimensions; (**h**) Hat core composite sandwich structure subjected to compression test; (**i**) Hat core composite sandwich structure subjected to bending test.

**Figure 2 polymers-14-02923-f002:**
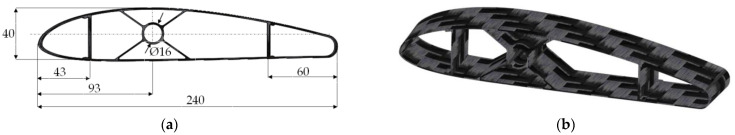
Wing sections: (**a**) front view of the wing section (mm); (**b**) internal composite structure of the wing segment subjected to three-point bending.

**Figure 3 polymers-14-02923-f003:**
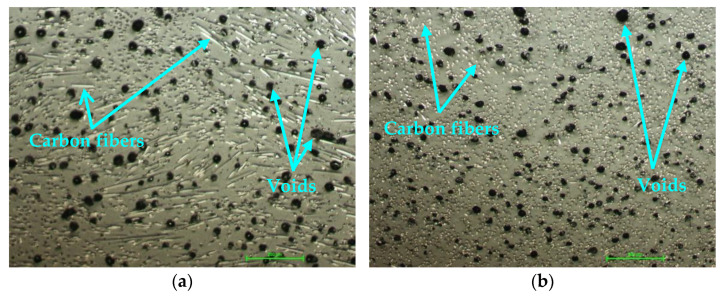
Microscopic analysis of the filament before 3D printing (100× magnification): (**a**) Longitudinal section of the filament; (**b**) Cross section of the filament.

**Figure 4 polymers-14-02923-f004:**
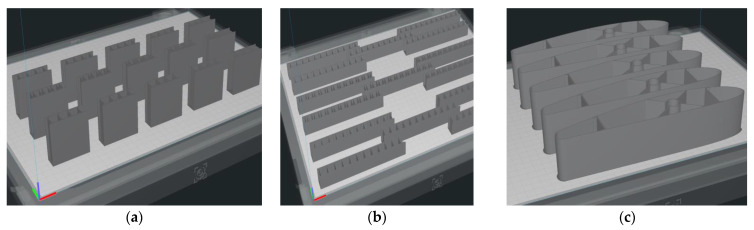
Establishing the 3D printing position of structures: (**a**) Composite sandwich specimens subjected to compression test; (**b**) Composite sandwich specimens subjected to three-point bending test; (**c**) Wing sections tested for three-point bending.

**Figure 5 polymers-14-02923-f005:**
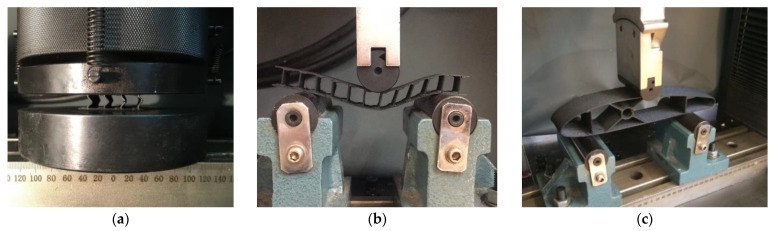
Testing of composite structures: (**a**) Flatwise compression of composite sandwich structures; (**b**) Three-point bending of composite sandwich structures; (**c**) Three-point bending of the wing sections.

**Figure 6 polymers-14-02923-f006:**
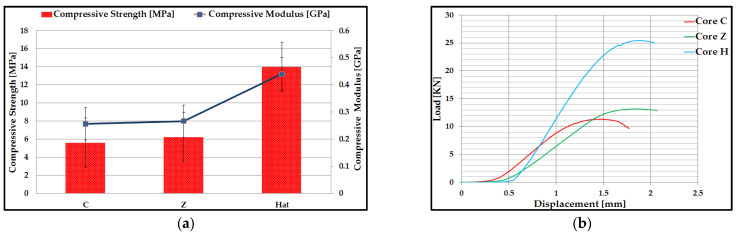
Compression test results of composite sandwich structures: (**a**) Mean values of compressive strength and modulus of elasticity; (**b**) Load–Displacement plot.

**Figure 7 polymers-14-02923-f007:**
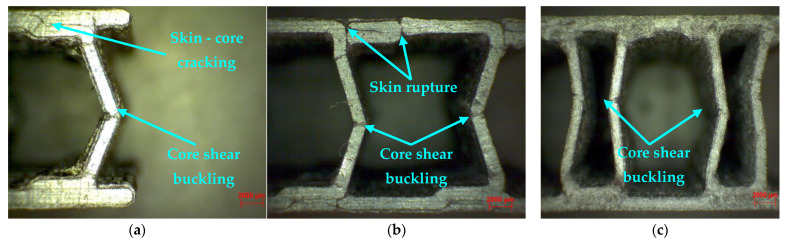
Microscopic analysis of sandwich structures subjected to flatwise compression (20× magnification): (**a**) C core; (**b**) Z core; (**c**) Hat core.

**Figure 8 polymers-14-02923-f008:**
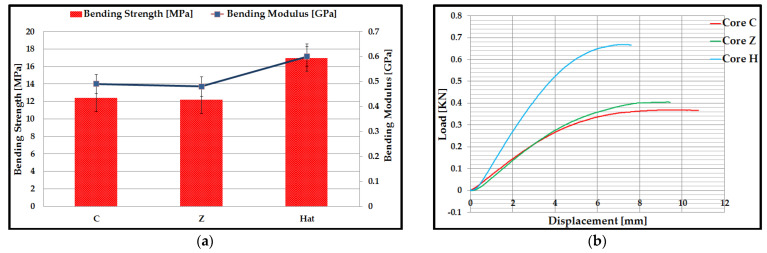
Results of three-point bending tests for sandwich structures: (**a**) Mean values of compressive strength and modulus of elasticity; (**b**) Load–Displacement curves.

**Figure 9 polymers-14-02923-f009:**
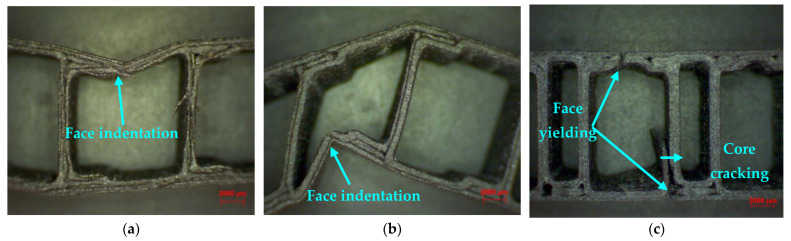
Microscopic analysis of sandwich structures tested for bending at three points (20X magnification): (**a**) C core; (**b**) Z core; (**c**) Hat core.

**Figure 10 polymers-14-02923-f010:**
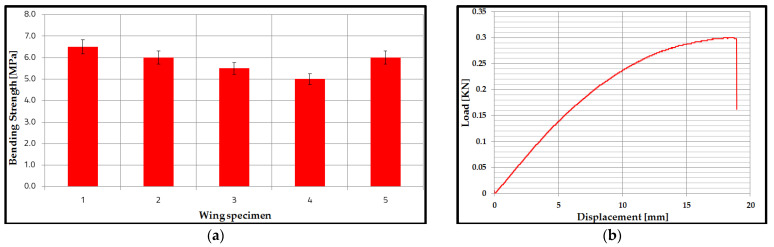
Results of three-point bending tests of the wing sections: (**a**) Mean values of bending strength; (**b**) Load Curve—Displacement.

**Figure 11 polymers-14-02923-f011:**
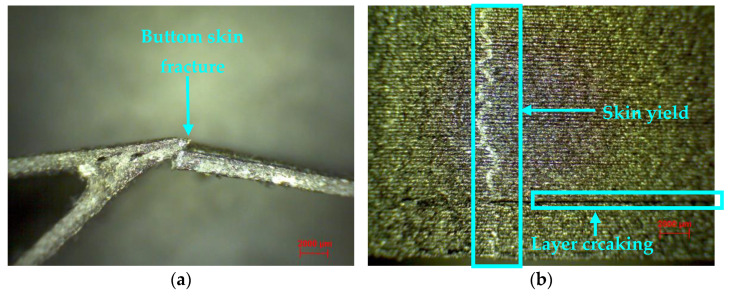
Microscopic analysis of three-point bending wing sections (magnification 20×): (**a**) Fracture of the wing skin; (**b**) Failure of the wing skin—top view.

**Figure 12 polymers-14-02923-f012:**
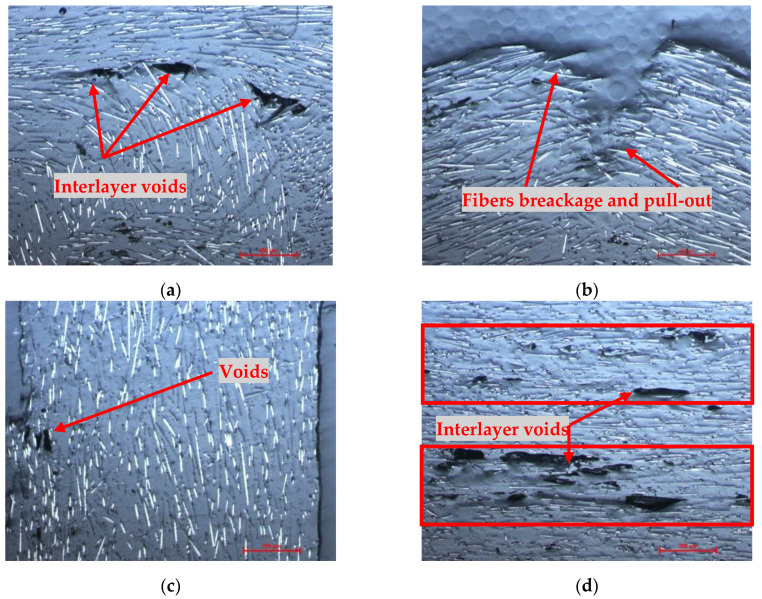
Microscopic analysis of wing sections tested for three-point bending (magnification 100×): (**a**–**c**) Longitudinal section; (**d**) Cross-section.

**Figure 13 polymers-14-02923-f013:**
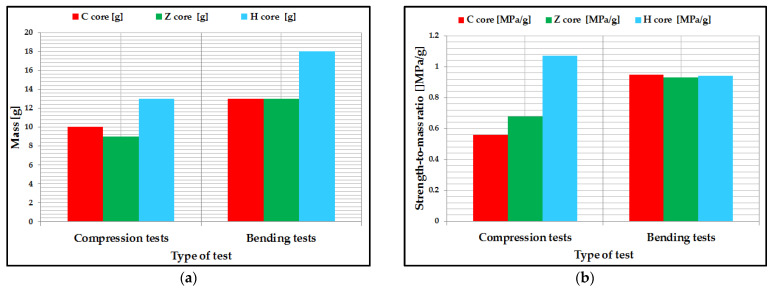
Analysis of the strength-to-mass ratio of composite sandwich specimens: (**a**) Mean mass of specimens; (**b**) The specific ratio for the six types of specimens tested.

**Figure 14 polymers-14-02923-f014:**
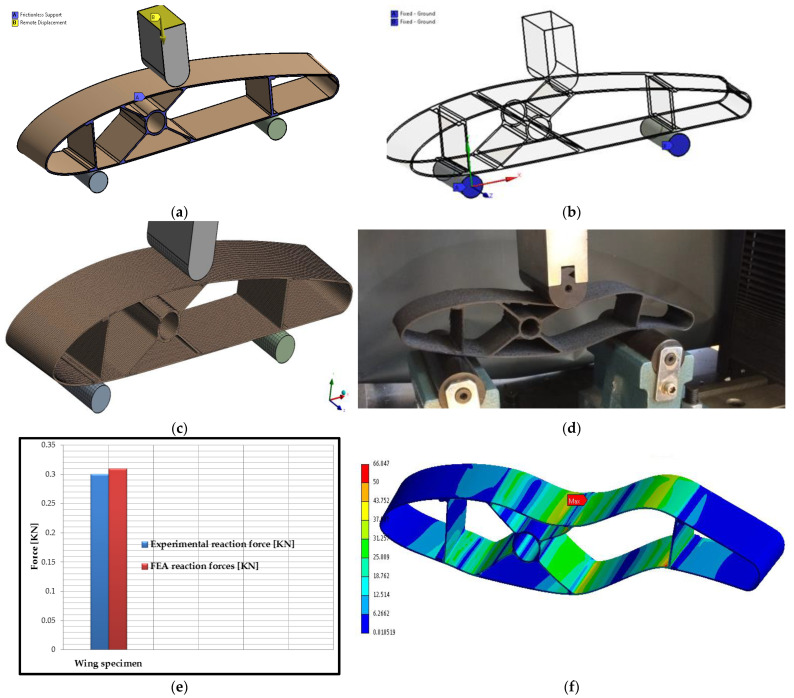
Finite element analysis of wing sections: (**a**,**b**) Determination of boundary conditions; (**c**) Model discretization; (**d**) Failure of the wing section following three-point bending tests; (**e**) Comparative analysis of reaction force; (**f**) Equivalent stress distribution [MPa].

**Table 1 polymers-14-02923-t001:** Parameters used in the manufacture of composite structures.

FFF Parameters	Value
Infill density [%]	100
Layer height [mm]	0.2
Printing speed [mm/sec]	50
Extrusion temperature [°C]	260
Bed Temperature [°C]	100
Nozzle diameter [mm]	0.6

**Table 2 polymers-14-02923-t002:** Statistical indicators calculated from flatwise compression tests of composite sandwich specimens.

Composite Sandwich Specimens	Mean (m)	Standard Deviation (s)	Coefficient of Variation (CV)%
C core—Compressive Strength (MPa)	5.60	0.55	9.78
Z core—Compressive Strength (MPa)	6.20	0.45	7.21
Hat core—Compressive Strength (MPa)	14.00	1.22	8.75
C core—Compressive Modulus (GPa)	0.25	0.02	8.99
Z core—Compressive Modulus (GPa)	0.26	0.02	8.00
Hat core—Compressive Modulus (GPa)	0.44	0.03	6.81

**Table 3 polymers-14-02923-t003:** Statistical indicators calculated from three-point bending tests of composite sandwich specimens.

Composite Sandwich Specimens	Mean (m)	Standard Deviation (s)	Coefficient of Variation (CV)%
C core—Bending Strength (MPa)	12.40	0.55	4.43
Z core—Bending Strength (MPa)	12.20	1.10	9.01
Hat core—Bending Strength (MPa)	17.00	0.71	4.17
C core—Bending Modulus (GPa)	0.49	0.01	2.04
Z core—Bending Modulus (GPa)	0.48	0.02	4.16
Hat core—Bending Modulus (GPa)	0.60	0.01	1.66

## Data Availability

Not applicable.

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
