# Peer review of "Compression and Bending Properties of Short Carbon Fiber Reinforced Polymers Sandwich Structures Produced via Fused Filament Fabrication Process"

_polymers, 2022, doi:10.3390/polym14142923_

Round 1

Reviewer 1 Report

The paper presents the design and fabrication by Fused Filament Fabrication (FFF) of composite sandwich structures with short fibers. The mechanical properites of core types C, Z and H were tested. The topic is very interesting, and a useful contribution to existing knowledge. I The following comments are provided to improve the current state of the paper. As an overall and based on the aforementioned comments the paper is worthy of publishing. The following comments are provided to improve the current state of the paper.

1.     Line 195, the citation is wrong.

2.     The unit should be given in figures, e.g., figure 1, figure 2 and so on.

3.     The basic information should be given, e.g., porosity of the specimens. The effects of the porosity on the mechanical properties of specimens is significant.

4.     In the section of “Flatwise compression performance of carbon fiber sandwich structures”, how did author determine the elastic modulus? Figure 6b shows the nolinear curve of the load-displacement, and how did author deal with the the initial line?

5.     How did author judge the failure of the specimens with applied load? Is it impossible to apply load? Or did it the local failure occur? Figure 9 shows the local failure of specimens is obvious.

6.     More information of FEM analysis should be given. The comparisons of the failure mode of the tested and simulation is useful. How did author judge the completion of the numerical calculation?

Reviewer 2 Report

This paper investigates compression and bending performance of composite sandwich structures fabricated by fused filament fabrication (FFF) of  with short fibers. My recommendation is accept after minor revision. Here are my comments:

(1) For comparison purposes, authors should consider whether to guarantee equivalent mass for composite sandwich structures with different cores.

(2) Authors should offer the mechanical properties (tensile, shear, etc.) of the specimens manufactured by FFF process.

(3) Authors should provide more information about numerical modeling (what kind of material model, what type of failure criterion, etc.. )
